# RETHINKING THE IMPACT OF HETEROGENEOUS SUB-LAYERS IN TRANSFORMERS

## ABSTRACT

Large language models (LLMs) using transformers have achieved state-of-the-art performance across a wide array of tasks. However, the sheer size and complexity of these models present both theoretical and practical challenges, *e.g.*, interpretation of the model behavior and deployment of edge devices. In this work, we revisit the architecture of transformers and propose a more granular understanding of the impacts of individual sublayers, *i.e.*, Multi-Head Attention (MHA) and Feed-Forward Network (FFN). We introduce a novel metric, *normalized relative impact factor*, that allows for progressive, heterogeneous layer pruning. This metric calculates the relative impact factor of each sublayer on the overall performance, normalized by the number of parameters. Our experiments demonstrate that our approach can lead to a 20% reduction in parameters and a 37% inference speedup, while maintaining minimal performance loss.

## 1 INTRODUCTION

Large Language Models (LLMs) (Brown et al., 2020; Zhang et al., 2022; Chowdhery et al., 2022; Touvron et al., 2023a;b; Dubey et al., 2024) using transformers have revolutionized natural language processing, achieving unprecedented successes in a variety of tasks such as multi-turn dialogue and long-context document understanding (Hadi et al., 2023; Zhao et al., 2023; Minaee et al., 2024). However, as the size of these models grows, they become increasingly difficult to deploy in resource-constrained environments, and their black-box nature makes them hard to interpret. On the one hand, the scale of LLMs, *e.g.*, even thousands of billion parameters in a single dense model (Zhang et al., 2022; Touvron et al., 2023a;b; Dubey et al., 2024), makes it difficult to interpret the behavior of the models. On the other hand, the increased memory consumption and computational demands have challenged the infrastructure of the deployment, particularly on resource-constrained edge devices. These challenges drive the need for techniques that can both explain and streamline transformer models without sacrificing their performance.

A central hypothesis for interpreting transformers is that their intermediate layers may operate in a common representation space (Sun et al., 2024). It suggests that residual connections (He et al., 2016), which function as identity mappings, enable information to flow from any layers directly to any subsequent layers (Gromov et al., 2024). We introduce a triangle-shaped unraveled view of transformer architecture in Figure 1(a), which clearly demonstrates the information flow. This consistent flow of information ties together the feature spaces of the different layers, resulting in a unified representation space. This perspective offers a lens to evaluate the impact of individual layers within the model (Sun et al., 2024; Freiberger et al., 2024; Schuster et al., 2022; Din et al., 2023; Varshney et al., 2023; Chen et al., 2023) or identify the redundant layers that contribute little to the overall performance (Song et al., 2024; Gromov et al., 2024; Del Corro et al., 2023). For example, as seen in Figure 1(b), an individual layer can hardly alter the topology of whole identity mappings, therefore it is reasonable that the model is robust to layer pruning. Empirical evidence supports this hypothesis. As illustrated in Figure 2(a), the L2 norm of the residual branches increases significantly in deeper layers, indicating that individual layers have a diminishing influence on the overall feature representation. Additionally, high cosine similarities between layers and the minimal performance loss observed when pruning certain layers further highlight the redundancy of many transformer layers, as illustrated in Figure 2(b) and Figure 2(c).

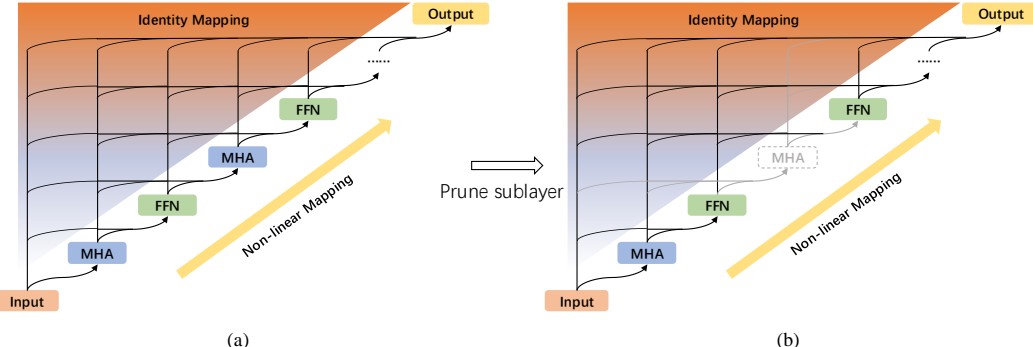

(a)                                                    (b)

Figure 1: The unraveled view of transformer architecture. (a) Transformer has a large scale of identity mappings. These identity mappings offer consistent information flow throughout the network and thus lead to a common representation space. (b) Prune one layer from the transformer has little impact on the topology, thus it is reasonable that the model is robust towards such distortion.

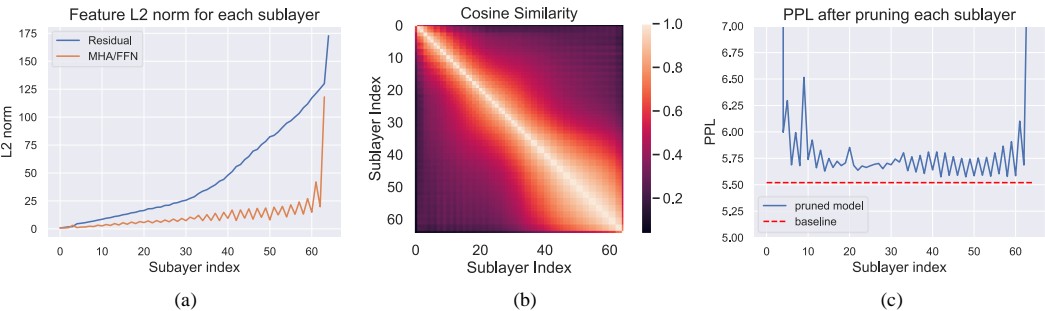

(a)                                  (b)                                  (c)

Figure 2: Illustration of common representation space. (a) The L2 norm of residual branches increases as the layer goes deeper, while the scale of the non-linear sublayers remains pretty much the same except for the last one. (b) Cosine similarity between each pair of sublayer features. High cosine similarity indicates that it is highly probable that these layers share a common representation space. (c) Perplexity on WikiText-2 dataset after pruning each sublayer. The minimal performance loss indicates the redundancy of the corresponding layer.

While prior research has typically treated entire transformer blocks as their unit of analysis (Sun et al., 2024; Song et al., 2024), we argue that a finer-grained approach is necessary to unveil the inherent importance of sublayers. A transformer block consists of two main components: the Multi-Head Attention (MHA) sublayer and the Feed-Forward Network (FFN) sublayer [1]. These two sublayers serve distinct functions within the architecture, and their contributions to the overall model performance are likely unequal. By dissecting the transformer block into its constituent sublayers, we can introduce a new perspective on layer impact factor that enables more nuanced pruning.

In this paper, we propose a heterogeneous layer pruning strategy based on a novel metric called **relative impact factor**, which quantifies the importance of each layer over the whole network. Using this metric, we derive a set of guidelines for recognizing the importance of heterogeneous layers by model size, layer position, layer type, and layer interaction. Upon the above analysis, we propose our progressive pruning approach, which reduces model complexity while preserving performance. Our experimental results demonstrate that our method achieves 20% parameter reduction and 37% speedups with minimal impact on perplexity and zero-shot accuracies, all without the need for fine-tuning. We hope our findings can help to understand the intrinsic mechanism of transformers.

---

[1]For simplicity, LayerNorm or RMSNorm is viewed as a part of the MHA / FFN module.This module is encompassed by the residual connection in recent architecture (Brown et al., 2020; Zhang et al., 2022; Chowdhery et al., 2022; Touvron et al., 2023a;b; Dubey et al., 2024).

## 2 RELATED WORKS

### 2.1 COMMON REPRESENTATION SPACE

Common representation space hypothesizes that the intermediate layers within a transformer network share the unified representation space. A notable technique in this context is the early exit (Schuster et al., 2022; Din et al., 2023; Varshney et al., 2023; Chen et al., 2023). These methods dynamically assess whether to process subsequent transformer blocks. When the model reaches a certain level of confidence in its early outputs, it can stop further processing, effectively reducing computational costs. Another recent approach proposes an alternative method to bypass transformer blocks (Del Corro et al., 2023). This strategy suggests that removing blocks, particularly those at the beginning of LLMs, might be more feasible. Layer shuffling is another way to shuffle transformer layers during training and inference to enhance the robustness of transformers (Freiberger et al., 2024). All these research are motivated by the same hypothesis that transformer blocks share the same representation space, therefore layer operations will not tremendously alter the representation feature of the network. Nevertheless, existing strategies primarily concentrate on the entire transformer block, neglecting a thorough examination of the efficacy of MHAs and FFNs. To assess the impact of these heterogeneous layers, we introduce a novel metric to quantify their importance, and subsequently, elucidate the relationships between importance and layer position, type, and interaction.

### 2.2 NETWORK PRUNING

Network pruning is one of the ways to compress the parameter size and accelerate model inference (LeCun et al., 1989; Hassibi et al., 1993; Han et al., 2015). Unstructured pruning targets the removal of individual weights, leading to sparse weight matrices within the model (Frantar & Alistarh, 2022; Sun et al., 2023; Frantar & Alistarh, 2023; Zhang et al., 2024). This sparsity is powerful in compressing parameters, whereas the complex sparse data access patterns hinder the model's acceleration. This complexity becomes particularly evident when using modern GPU hardware, as these systems are typically optimized for dense matrix operations (Wang, 2020; Shi et al., 2020). Structured pruning, on the other hand, involves the elimination of predefined units of weights to create more hardware-friendly patterns. 2:4 pruning uses a semi-structured approach to eliminate weights to create a more efficient pattern (Sun et al., 2023; Frantar & Alistarh, 2023; Zhang et al., 2024; Mishra et al., 2021). Channel pruning focuses on eliminating entire channels, thus preserving the pruned weight matrix's dense nature (Ashkboos et al., 2024; Ma et al., 2023). Notably, the channel pruning ratio does not necessarily translate to a proportional improvement in the end-to-end inference speed of LLMs. This limitation arises because transformer blocks in LLMS encompass various operations beyond matrix multiplication such as layer normalization and self-attention.

Layer pruning, which drops the entire layer, has a substantial speedup on common devices (Song et al., 2024; Men et al., 2024). Previous layer pruning methods treat the transformer layer as a homogeneous unit, pruning both the multi-head attention (MHA) and feed-forward network (FFN) together (Song et al., 2024). However, the MHA and FFN serve distinct roles, with MHA focusing on token mixing and FFN on channel mixing, each contributing unique information to the model. These components are not inherently coupled, and thus, they should be pruned independently to preserve their distinct functionalities. Furthermore, MHA and FFN also have different numbers of parameters, making it challenging to determine which component is more critical for constructing an efficient network.. In our paper, we propose a new metric to measure the importance of sublayers and then propose a new method of layer pruning.

## 3 LESION STUDY ON THE IMPORTANCE OF HETEROGENEOUS LAYERS

The critical question about layer pruning is: *Is there a way to figure out the importance of heterogeneous layers in transformers?* In this section, we first introduce a metric named Relative Impact factor (RI) to quantify the importance of the layers. Then we outline 4 guidelines for understanding the importance of heterogeneous layers in transformers.

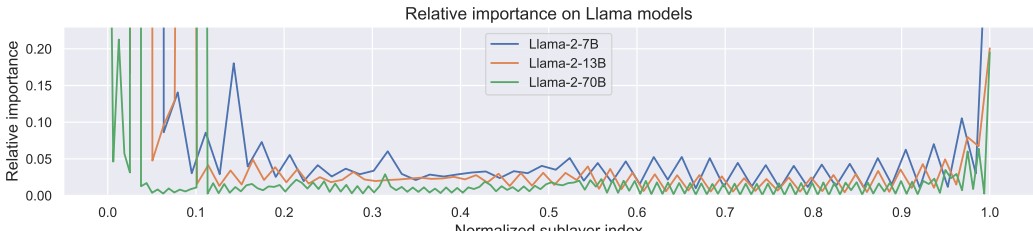

Figure 3: Relative impact factor on model size and layer position.

## 3.1 RELATIVE IMPACT FACTOR

We introduce Relative Impact factor (RI) as a metric to quantify the contribution of individual transformer layers. Unlike previous metrics, which assess the importance of entire blocks, our metric focuses on the contributions of MHA and FFN sublayers separately. Following the training and evaluation strategies of LLMs, perplexity is one of the common metrics for quantifying the model performance. Therefore, we define the relative impact factor of a layer is computed by evaluating the change in model perplexity when the layer is lesioned (*i.e.*, pruned or disabled):

$$\mathrm{RI}(M_i|M) = \frac{\mathrm{PPL}(M - M_i) - \mathrm{PPL}(M)}{\mathrm{PPL}(M)}, \tag{1}$$

where $M$ denotes the baseline model, $M_i$ denotes layer $i$ of model $M$, $M - M_i$ denotes prune layer $M_i$ from the $M$. This metric measures the increase in relative perplexity by pruning one layer from the baseline model, thereby revealing the layer's contribution to the model's predictive performance.. A key advantage of RI is its flexibility in comparing different baseline models, such as those with varying architectures or parameter sizes. A higher RI value suggests that the layer's removal would lead to a significant deterioration in model performance, implying that the layer plays a more critical role in the model's overall performance. RI is processed with the inference-only approach without any fine-tuning, aligning with contemporary practices in the post-training processing of LLMs. For PPL calculation, we randomly select 128 samples with length 2048 from the WikiText-2 (Merity et al., 2016) training dataset as evaluation data, following the approach used in previous works (Ashkboos et al., 2024).

## 3.2 GUIDELINES FOR HETEROGENEOUS LAYER IMPORTANCE

With the relative impact factor proposed above, we conduct extensive experiments and derive several key observations regarding the importance of heterogeneous layers in transformers. Since different model sizes are composed of different numbers of layers, we use min-max normalization to normalize the sublayer index. For a transformer network with $N$ transformer block, we have $2N$ sublayers, and each index $i \in \{0, 1, \ldots, 2N - 1\}$ is normalized to $i/(2N - 1)$. In this way, we can compare the models with different parameter sizes.

**Guideline 1: Intermediate layers are redundant.** The relative impact factor of the Llama-2 series, illustrated in Figure 3, shows pruning individual intermediate layers results in minimal performance degradation. It reveals a significant degree of redundancy in the middle layers of transformer models. This finding aligns with the hypothesis that these layers operate within a shared representation space. Consequently, removing one layer has little effect on the topology of identity mappings, as shown in Figure 1(b).

In contrast, sublayers located near the input or output exhibit much higher importance due to their proximity to the input embeddings and output classifiers, suggesting that their representation space differs from that of intermediate features. This conclusion is further supported by Figure 2(a), where we observe that the residual branch and the MHA/FFN branch in the input layers have similar L2 norms, indicating an adequate contribution to the branch. As we progress deeper into the network, the L2 norm of the MHA/FFN branches becomes relatively lower due to the increasing residual branch. Notably, there is a sudden increase in the L2 norm at the last sublayer, indicating a resurgence in its relative impact factor.

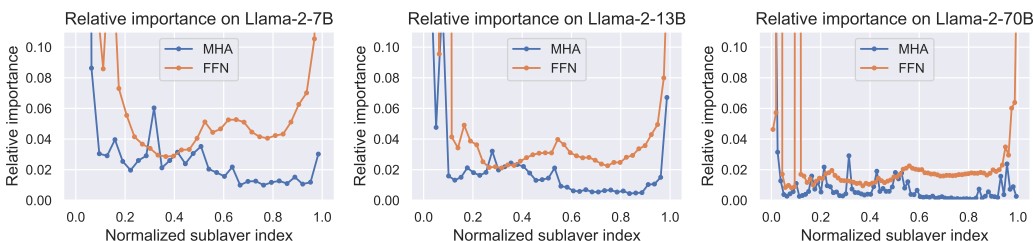

Figure 4: Relative impact factor on layer type.

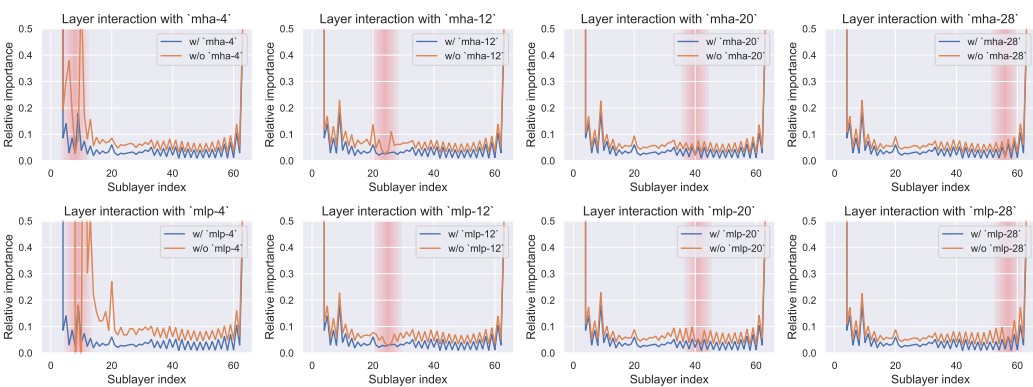

Figure 5: Layer interaction within the model. The red span indicates the surroundings of the pruned sublayer. Pruning shallow layers will prominently affect the importance of adjacent layers, while deep layers are more robust to layer pruning.

**Guideline 2: Large models have more layer redundancy.** In larger transformer models, redundancy among layers tends to increase, allowing for more aggressive pruning with minimal impact on performance. As depicted in Figure 3, larger models show lower relative impact factor across most sublayers, and pruning these layers results in negligible performance loss. This suggests that larger models possess a greater degree of overparameterization.

**Guideline 3: MHAs are more redundant than FFNs.** The type of layer also plays a crucial role in determining relative impact factor. As shown in Figure 4, MHA sublayers generally exhibit higher redundancy than FFN sublayers. Several factors contribute to this observation. Firstly, MHAs typically contain fewer parameters than FFNs. For instance, the Llama MLP possesses twice the number of parameters compared to the MHA layers. As a result, pruning one MHA generally leads to a smaller increase in PPL than pruning one MLP. Secondly, the L2 norm of the MHA branches is smaller than the FFN branches, as illustrated in Figure 2(a), indicating that MHAs exert less influence on the main feature branch. Lastly, MHAs are susceptible to the rank collapse problem, which leads to similar outputs for different tokens, as reported by Dong et al. (2021). Consequently, deeper MHA layers contributes less valuable information to overall model performance.

**Guideline 4: Layer interaction affects importance.** The interaction between sublayers significantly influences their relative importance. To assess these interactions, we conducted a two-stage experiment using relative impact factor scores. Initially, we evaluated the relative impact factor of each sublayer within the complete model. Subsequently, we pruned one sublayer and re-evaluated the importance of the remaining sublayers. Given that our metric evaluates sublayer importance relative to the entire network, changes in importance after pruning are expected. As illustrated in Figure 5, the removal of a sublayer leads to changes in the relative impact factors. Notably, pruning a shallow sublayer can result in substantial shifts in layer importance, highlighting the strong interdependencies between shallow layers. These results imply that one-shot layer pruning methods (Men

et al., 2024), which are incapable of dealing with layer interaction, cannot perform optimal pruning. Layer interaction highlights the necessity for a careful and interdependent pruning strategy.

# 4 HETEROGENEOUS LAYER PRUNING

Our analysis highlights the redundancy of sublayers in large language models (LLMs), necessitating the need to prune these redundant layers to improve model efficiency. To develop an effective pruning method, we must address two key questions: *Which sublayers are most suitable for pruning?* and *How can we prune multiple sublayers effectively?* For the first question, relative impact factor offers a potential solution, it does not account for the number of parameters in each layer. Therefore, we propose a metric called normalized relative impact factor, which calibrates relative impact factor based on the parameter count of each sublayer, allowing us to prune layers that contribute the least per parameter. This approach yields a more efficient and streamlined model. For the second question, traditional one-shot methods fail to consider interactions between sublayers. To overcome this limitation, we adopt a progressive search approach that iteratively identifies and prunes the least important layers. The details of our method are outlined below:

## 4.1 NORMALIZED RELATIVE IMPACT FACTOR

Previous layer pruning methods have typically utilized perplexity (PPL) or cosine similarity as metrics. However, these metrics are not well-suited for heterogeneous layer pruning, where layers vary significantly in their parameter counts. For instance, the Llama MLP possesses twice the number of parameters compared to the MHA layers. As a result, pruning one MHA generally leads to a smaller increase in PPL than pruning one MLP. To maintain a consistent pruning ratio across the network, more MHA layers would need to be removed, which could inadvertently lead to greater performance degradation.

To accurately assess the importance of each layer, we normalize the RI metric by the number of parameters in each sublayer. This normalization allows for a more precise evaluation of layer significance, accounting for the varying sizes of sublayers. For each layer $i$, we calculate the normalized relative impact (NRI) metric as follows::

$$\text{NRI}(M_i|M) = \frac{\text{RI}(M_i|M)}{\text{Params}(M_i)}. \tag{2}$$

This metric represents the RI per parameter, enabling a fair comparison among sublayers of different sizes. It ensures that larger sublayers, which inherently have a higher parameter count, do not disproportionately skew the overall importance score. Consequently, it enables more effective pruning, maintaining parameter efficiency while reducing redundancy. Thus we can optimize the model's efficiency with little performance compromise.

## 4.2 PROGRESSIVE SEARCH

As illustrated in Algorithm 1, our pruning method employs a progressive search strategy that iteratively removes layers based on their NRI. In each iteration, we calculate the NRI metric for all sublayers, enabling a comprehensive evaluation of their contributions.. We then identify and prune the least important layer, thereby minimizes the impact on model performance. This iterative process is repeated until the desired model compression ratio is attained.

The primary benefit of progressive pruning lies in its capacity to capture layer interdependencies. Through iterative evaluation of each layer's importance, we can quantify the effects of layer removal on the relative significance of the remaining layers. This dynamic evaluation allows for a more informed pruning process, ensuring that critical layers are retained while redundant ones are eliminated. Consequently, this method yields an efficient model with minimal performance degradation. Overall, the progressive search strategy facilitates a more nuanced and effective pruning process, ultimately leading to a streamlined model that maintains its predictive capabilities.

---

**Algorithm 1** Progressive layer search

---

**Require:** Base model $M$, number of layers to prune $N$, calibration set $D$

 Evaluate model $M$ and get PPL $l$            ▷ Get baseline performance
 **while** $N \geq 0$ **do**                     ▷ Progressive search
  **for** each layer $i$ **do**
   Evaluate model $\{M - M_i\}$ on calibration set $D$ and get PPL $l_i$
   Count the number of parameters of sublayer $M_i$ as $p_i$
   Calculate normalized relative impact factor $\mathrm{NRI}[i] = (l_i - l)/lp_i$    ▷ Parameter normalized
  **end for**
  Sort NRI by index
  Identify smallest index $i$ in NRI
  $M \leftarrow \{M - M_i\}$                ▷ Prune the least important layer
  $N \leftarrow N - 1$
  $l \leftarrow l_i$                  ▷ Update baseline model performance
 **end while**

---

Figure 6: Location and layer type of pruned sublayers after applying our method with target sparsity of 20%.

## 5 EXPERIMENTS

### 5.1 EXPERIMENTAL SETUP

We implement our method using PyTorch (Paszke et al., 2019), leveraging the HuggingFace Transformers library (Wolf et al., 2020) for model manipulation and execution. All pruning experiments are conducted on NVIDIA A100 GPUs with 80GB of memory. The progressive search method relies on 128 randomly selected samples from the WikiText-2 Merity et al. (2016) training dataset as calibration data, following the approach adopted in previous works (Song et al., 2024). Although our method employs a detailed pruning strategy, the total execution time remains comparable to other state-of-the-art pruning methods (see Appendix A.2 for more details).

Our evaluation covers models from both the OPT (Zhang et al., 2022) and LLaMA-2 (Touvron et al., 2023b) families. We assess them under two target sparsity levels: 10% and 20%, following the experimental setups used in prior research (Song et al., 2024). We compare the performance of our method with several other pruning techniques, including SparseGPT (Frantar & Alistarh, 2023), Wanda (Sun et al., 2023), and DSnoT Zhang et al. (2024), which utilize 2:4 structured pruning, as well as LLM-Pruner Ma et al. (2023) and SliceGPT Ashkboos et al. (2024), which adopt channel-wise pruning. In addition, we benchmark our method against SLEB (Song et al., 2024), which prunes entire transformer blocks, providing a more coarse-grained approach to model optimization.

### 5.2 PRUNED SUBLAYERS

We illustrate the pruned sublayers in Figure 6. The locations and types of pruned layers vary significantly across the target models. In LLaMA-2 models, middle and later sublayers are removed more frequently, whereas earlier sublayers are often pruned in OPT-13B. This indicates that sensitivity to remove sublayers depends on the specific model architecture. Furthermore, our findings reveal that MHAs are more likely to be pruned in LLaMA-2 models, suggesting that these layers exhibit greater redundancy within this architecture. In contrast, OPT models show no clear preference for either layer type, highlighting the variability in layer importance across different transformer models.

Table 1: Perplexity results on C4 dataset and throughput (tokens/s) results. We measure throughput of each method with LLaMA-2-70B on 2 NVIDIA A100 GPUs.

| Method | Pruning Unit | Sparsity | Tokens/s | Throughput Improve. | OPT | | | | LLaMA-2 | | |
|---|---|---|---|---|---|---|---|---|---|---|---|
| | | | | | 6.7B | 13B | 30B | 66B | 7B | 13B | 70B |
| Dense | - | 0% | 299 | 1.00× | 12.71 | 12.06 | 11.44 | 10.99 | 7.26 | 6.3 | 5.71 |
| SparseGPT | 2:4 | 50% | 293 | 0.98× | 16.42 | 14.85 | 13.17 | 12.25 | 13.54 | 11.39 | 8.16 |
| Wanda | 2:4 | 50% | 293 | 0.98× | 19.03 | 16.18 | 16.18 | 8414.05 | 15.57 | 12.47 | 8.10 |
| DSnoT | 2:4 | 50% | 293 | 0.98× | 18.41 | 16.51 | 14.71 | 8360.81 | 15.56 | 12.22 | 8.15 |
| LLM-Pruner | Channel | 20% | 314 | 1.05× | - | - | - | - | 12.25 | 10.43 | - |
| SliceGPT | Channel | 20% | 314 | 1.05× | 23.76 | 17.49 | 13.38 | 11.80 | 26.06 | 22.90 | 15.84 |
| SliceGPT | Channel | 25% | 331 | 1.11× | 27.35 | 19.43 | 14.46 | 12.29 | 32.74 | 29.86 | 20.03 |
| SliceGPT | Channel | 30% | 343 | 1.15× | 33.43 | 22.58 | 15.89 | 13.08 | 41.69 | 38.43 | 25.79 |
| SLEB | Block | 20% | 381 | 1.27× | 15.99 | 13.81 | 12.74 | 12.54 | 12.32 | 9.42 | 7.31 |
| Ours | Sublayer | 10% | 359 | 1.20× | 13.82 | 12.41 | 12.07 | 11.22 | 8.76 | 7.77 | 6.22 |
| Ours | Sublayer | 20% | 410 | 1.37× | 15.80 | 13.77 | 13.01 | 12.50 | 10.91 | 9.11 | 7.30 |

Table 2: Mean accuracies (%) on zero-shot tasks and latency results. We measure latency of each method with LLaMA-2-70B on 2 NVIDIA A100 GPUs.

| Method | Pruning Unit | Sparsity | Latency(ms) | Speedup | OPT | | | | LLaMA-2 | | |
|---|---|---|---|---|---|---|---|---|---|---|---|
| | | | | | 6.7B | 13B | 30B | 66B | 7B | 13B | 70B |
| Dense | - | 0% | 1718.4 | 1.00× | 60.70 | 61.79 | 64.40 | 66.16 | 69.00 | 71.76 | 76.57 |
| SparseGPT | 2:4 | 50% | 1555.5 | 1.10× | 54.94 | 56.76 | 59.96 | 62.33 | 58.23 | 63.06 | 71.87 |
| Wanda | 2:4 | 50% | 1555.5 | 1.10× | 53.14 | 55.12 | 58.89 | 35.93 | 55.59 | 61.23 | 72.34 |
| SliceGPT | Channel | 20% | 1658.7 | 1.04× | 56.31 | 60.20 | 63.65 | 65.74 | 58.17 | 63.45 | 72.34 |
| SliceGPT | Channel | 25% | 1440.7 | 1.19× | 54.28 | 59.27 | 62.11 | 65.17 | 55.49 | 58.90 | 69.75 |
| SliceGPT | Channel | 30% | 1364.2 | 1.26× | 53.00 | 57.42 | 61.27 | 64.24 | 51.50 | 55.16 | 66.11 |
| SLEB | Block | 20% | 1364.1 | 1.26× | 57.61 | 60.08 | 62.86 | 62.53 | 56.80 | 62.96 | 70.81 |
| Ours | Sublayers | 10% | 1432.0 | 1.20× | 60.01 | 61.65 | 64.08 | 65.76 | 64.45 | 69.96 | 75.18 |
| Ours | Sublayers | 20% | 1253.9 | 1.37× | 57.69 | 60.77 | 63.22 | 64.80 | 60.26 | 65.09 | 72.80 |

## 5.3 LANGUAGE MODELING

We assess the linguistic capabilities of our pruned LLMs through standard language modeling tasks, measuring their performance on the C4 (Raffel et al., 2020) validation dataset, a widely-used corpus for evaluating language models. Table 1 presents the perplexity results. Perplexity is a key metric for evaluating language models, as it indicates how well a model predicts sequences of words. Despite achieving high levels of sparsity, our method shows minimal increases in perplexity, highlighting its ability to maintain model performance. Notably, our approach achieves considerable speedup, even when compared to models pruned at similar or higher sparsity levels. We discuss these speedup results in more detail in Section 5.5. Overall, our progressive pruning method demonstrates the ability to prune models effectively without a significant loss in language modeling capabilities.

## 5.4 ZERO-SHOT TASKS

To evaluate the generalization capabilities of our pruned models, we measure their performance on various zero-shot downstream tasks. Following the evaluation procedures outlined in previous research, we assess the models on PIQA (Bisk et al., 2020), WinoGrande (Sakaguchi et al., 2019), HellaSwag (Zellers et al., 2019), ARC-easy (Clark et al., 2018), and ARC-challenge (Clark et al., 2018), using the LM Evaluation Harness (Gao et al., 2023) with default settings. Table 2 presents the average accuracies for all zero-shot tasks.

Our method consistently outperforms other pruning methods, particularly at 10% sparsity, preserving the original performance of the model. Even on 20% sparsity, our approach continues to exceed other techniques, maintaining strong performance on most tasks. Detailed task-wise accuracies can

Table 3: LLaMA-2 latency for prompt processing on A100.

| Method | Sparsity | 7B | | | 13B | | | 70B | | |
|---|---|---|---|---|---|---|---|---|---|---|
| | | #GPU | Latency | Speedup | #GPU | Latency | Speedup | #GPU | Latency | Speedup |
| Dense | - | 1 | 240.0 | 1.00× | 1 | 397.3 | 1.00× | 2 | 1718.4 | 1.00× |
| 2:4 Pruning | 50% | 1 | 218.2 | 1.10× | 1 | 372.2 | 1.07× | 2 | 1555.5 | 1.10× |
| Ch. Pruning | 25% | 1 | 213.3 | 1.13× | 1 | 349.9 | 1.14× | 2 | 1440.7 | 1.19× |
| SLEB | 10% | 1 | 209.3 | 1.15× | 1 | 355.1 | 1.12× | 2 | 1529.1 | 1.12× |
| SLEB | 20% | 1 | 187.3 | 1.28× | 1 | 316.0 | 1.26× | 2 | 1364.1 | 1.26× |
| Ours | 10% | 1 | 207.5 | 1.16× | 1 | 330.5 | 1.20× | 2 | 1432.0 | 1.20× |
| Ours | 20% | 1 | 180.1 | 1.33× | 1 | 290.5 | 1.37× | 2 | 1253.9 | 1.37× |

be found in Appendix A.3. These results highlight the robustness of our method, demonstrating that it retains the model's generalization capabilities even under significant pruning.

## 5.5 SPEEDUP

One of the key advantages of our pruning method is the substantial speedup it provides during inference. We evaluate the inference latency of our pruned models in comparison to other pruning techniques, including 2:4 structured pruning, 25% channel-wise pruning, and whole transformer block pruning. All latency measurements are conducted using the HuggingFace Transformers implementation for each method.

Table 3 presents the latency results for LLaMA models. Our pruned models show substantial reductions in inference time, particularly due to the pruning of multi-head attention (MHA) layers, which are computationally expensive components of transformer models. The significant reduction in processing time makes our method highly effective for real-world applications where speed is crucial. In contrast, previous pruning methods, including structured and block-wise pruning, fail to achieve comparable improvements in speed. The efficiency gains we achieve underscore the effectiveness of our approach, making it not only a strong performer in terms of accuracy but also a practical solution for deploying large language models in latency-sensitive environments.

## 6 LIMITATION AND DISCUSSION

In this paper, we introduce a relative perspective to assess the importance of individual sublayers within transformer models. While perplexity serves as a standard measure for natural language processing, it may not be suitable for other modalities. For instance, vision or audio processing requires different metrics, such as cross-entropy for classification or reconstruction loss for generation. Regardless of the specific metric used, we emphasize the utility of relative measurements as a general framework for evaluating how much a layer contributes to overall model performance across various domains. It can offer a more adaptable and effective approach to recognizing the importance of sublayers.

Another consideration is that parameter normalization is not the only method available for adjusting the relative impact factor of sublayers. Alternative normalization methods can be applied depending on the specific needs of an application. For instance, if a task prioritizes reducing inference latency or memory cost, the latency or memory of each sublayer could be used for normalization instead of the number of parameters. In this case, layers with higher latency or memory consumption would be pruned more aggressively, such as MHA when dealing with long input contexts. This flexibility allows the pruning strategy to be tailored to the constraints of the deployment scenario, making it adaptable to a variety of use cases.

Finally, investigating the relative impact factor of layers could have broader implications beyond model pruning. For example, the insights gained from understanding layer redundancy can be applied to mixed-precision quantization, where more redundant layers could be quantized to lower precision without significant loss in performance (Frantar et al., 2023; Lin et al., 2023; Lee et al., 2024). Additionally, similar methods could be used to inform decisions about model compression

techniques, like low-rank factorization or parameter sharing, to optimize model efficiency. We leave these problems for future work.

## 7 CONCLUSION

In this paper, we extensively analyze the importance of heterogeneous layers in transformer-based LLMs. Building upon our analysis, we propose a novel pruning strategy for transformer models, which leverages the normalized relative impact factor to selectively eliminate redundant layers while maintaining performance. Our progressive pruning approach effectively captures layer interactions, resulting in negligible perplexity increases and substantial speedups, especially in zero-shot learning scenarios. This work provides insight into understanding heterogeneous layer behaviors and a practical method for optimizing large models without compromising their capabilities.

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

# A APPENDIX

## A.1 SELECTED SUBLAYERS

We summarize the indices of sublayers selected for removal in Table 4. The order of the indices presented in Table 4 corresponds to the order in which they are removed.

Table 4: Indices of removed sublayers when applying our method on different LLM models. In 20% sparsity scenarios, it initially removes sublayers selected for 10% sparsity and then selects additional sublayers for removal to reach the desired sparsity level.

| Type | Model Size | #Blocks | 10% Sparsity Removed Indices | Count | 20% Sparsity Removed Indices | Count |
|------|------|---------|-----------------|-------|-----------------|-------|
| OPT | 6.7B | 32 | MHA-6, MHA-4, FFN-5, MHA-3, FFN-7, MHA-8, MHA-29, FFN-13 | MHA:5 FFN: 3 | +MHA-14, MHA-31, MHA-30, MHA-18, FFN-17, FFN-28, MHA-25 | MHA:+5 FFN: +2 |
| | 13B | 40 | MHA-4, MHA-2, FFN-5, MHA-5, MHA-8, MHA-17, FFN-8, MHA-14, FFN-13 | MHA:6 FFN: 3 | +FFN-2, MHA-37, MHA-12, MHA-39, FFN-17, FFN-11, MHA-38, MHA-18, FFN-32 | MHA:+5 FFN: +4 |
| | 30B | 48 | FFN-4, MHA-5, MHA-6, FFN-6, MHA-14, FFN-2, MHA-2, MHA-16, FFN-13, MHA-13, FFN-15 | MHA:6 FFN: 5 | +MHA-20, FFN-20, MHA-45, MHA-11, FFN-11, MHA-47, FFN-28, FFN-14, MHA-46, MHA-29 | MHA:+6 FFN: +4 |
| | 66B | 64 | MHA-16, FFN-8, MHA-9, FFN-3, MHA-4, MHA-6, FFN-16, MHA-10, FFN-7, MHA-23, FFN-6, FFN-14, FFN-20 | MHA:6 FFN: 7 | +MHA-63, MHA-21, MHA-60, FFN-23, MHA-18, FFN-18, MHA-59, MHA-27, FFN-27, MHA-62, FFN-19, FFN-34, MHA-57, FFN-49 | MHA:+8 FFN: +6 |
| LLaMA-2 | 7B | 32 | MHA-24, MHA-27, MHA-21, MHA-30, FFN-11, MHA-11, FFN-10, MHA-12 | MHA:6 FFN: 2 | +MHA-25, FFN-12, MHA-13, FFN-26, MHA-23, FFN-9, MHA-10 | MHA:+4 FFN: +3 |
| | 13B | 40 | MHA-33, MHA-34, MHA-31, MHA-35, MHA-25, MHA-28, MHA-30, MHA-27, MHA-26, MHA-24, FFN-10, MHA-10 | MHA:11 FFN: 1 | +FFN-11, MHA-29, FFN-29, MHA-11, MHA-32, FFN-12, MHA-12, FFN-9 | MHA:+4 FFN: +4 |
| | 70B | 80 | MHA-65, MHA-61, MHA-59, MHA-64, MHA-62, MHA-60, MHA-63, MHA-57, MHA-68, FFN-6, MHA-53, MHA-58, MHA-49, FFN-29, MHA-66, MHA-54, MHA-79, FFN-31, MHA-30, FFN-27, MHA-73, FFN-13, MHA-14, MHA-31, FFN-30 | MHA:19 FFN: 6 | +MHA-32, FFN-55, FFN-28, MHA-29, FFN-69, MHA-71, FFN-22, MHA-23, FFN-54, MHA-56, FFN-70, FFN-33, MHA-28, FFN-34, MHA-35, MHA-34 | MHA:+8 FFN: +8 |

## A.2 RUN TIME OF PRUNING METHODS

We compare the execution time of our method against previous approaches for pruning LLaMA-2 models. Specifically, we measure the time required to identify and remove elements based on each pruning strategy. The results are shown in Table 5. Compared to SLEB which prunes entire transformer blocks, our approach operates at a finer granularity which prunes sublayers. This results in a longer pruning time due to the more detailed analysis and progressive search strategy.

Despite the run time during the pruning process, a key advantage of our method is that the pruning algorithm is applied only once. After the model is pruned, it requires no further iterations or re-applications of the pruning procedure. This one-time cost is outweighed by the significant long-term benefits: the pruned model offers tremendous speedup during inference and training, vastly improving efficiency over time. In practice, this means the initial investment in pruning time is quickly recouped by the model's enhanced performance, making our approach not only feasible but also advantageous for real-world applications.

Table 5: Comparison of runtime for pruning LLaMA-2 using NVIDIA A100 GPU

| Method | 7B | | | 13B | | | 70B | | |
|--------|------|----------|----------|------|----------|----------|------|----------|----------|
| | #GPU | Run Time (s) | Inference speedup | #GPU | Run Time (s) | Inference speedup | #GPU | Run Time (s) | Inference speedup |
| SparseGPT | 1 | 528 | 1.10× | 1 | 920 | 1.07× | 3 | 5040 | 1.10× |
| Wanda | 1 | 64 | 1.10× | 1 | 104 | 1.07× | 2 | 480 | 1.10× |
| SliceGPT | 1 | 452 | 1.13× | 1 | 633 | 1.14× | 1 | 3010 | 1.19× |
| SLEB 20% | 1 | 113 | 1.28× | 1 | 279 | 1.26× | 2 | 5420 | 1.26× |
| Ours 20% | 1 | 465 | 1.33× | 1 | 1132 | 1.37× | 2 | 22116 | 1.40× |

## A.3 ZERO-SHOT TASKS

We show the detailed accuracies on zero-shot tasks as shown in Table 6

Table 6: Accuracies (%) on zero-shot tasks

| Model | Method | Sparsity | PIQA | WinoGrande | HellaSwag | ARC-e | ARC-c | Avg. |
|-------|--------|----------|------|------------|-----------|-------|-------|------|
| OPT-6.7B | Dense | - | 76.39 | 65.19 | 67.16 | 60.14 | 34.64 | 60.70 |
| | SpareGPT | 2:4 (50%) | 74.21 | 60.77 | 57.25 | 53.03 | 29.44 | 54.94 |
| | Wanda | 2:4 (50%) | 71.76 | 60.22 | 54.20 | 51.18 | 28.33 | 53.14 |
| | SliceGPT | 25% | 70.35 | 60.62 | 58.15 | 52.78 | 29.52 | 54.28 |
| | SliceGPT | 30% | 68.61 | 60.69 | 54.56 | 52.15 | 29.01 | 53.00 |
| | SLEB | 20% | 74.92 | 61.33 | 62.13 | 57.07 | 32.59 | 57.61 |
| | Ours | 10% | 75.90 | 64.33 | 66.70 | 59.76 | 33.36 | 60.01 |
| | Ours | 20% | 74.10 | 62.67 | 63.32 | 55.09 | 33.28 | 57.69 |
| OPT-13B | Dense | - | 76.82 | 64.80 | 69.81 | 61.87 | 35.67 | 61.79 |
| | SpareGPT | 2:4 (50%) | 74.16 | 62.43 | 59.18 | 56.27 | 31.74 | 56.76 |
| | Wanda | 2:4 (50%) | 72.25 | 61.72 | 57.97 | 53.96 | 29.69 | 55.12 |
| | SliceGPT | 25% | 73.67 | 64.25 | 63.28 | 60.52 | 34.64 | 59.27 |
| | SliceGPT | 30% | 71.82 | 62.90 | 60.66 | 58.80 | 32.94 | 57.42 |
| | SLEB | 20% | 75.90 | 63.46 | 66.87 | 59.60 | 34.56 | 60.08 |
| | Ours | 10% | 76.44 | 63.06 | 69.80 | 62.77 | 36.18 | 61.65 |
| | Ours | 20% | 76.01 | 62.83 | 69.05 | 61.15 | 34.81 | 60.77 |
| OPT-30B | Dense | - | 78.07 | 68.19 | 72.27 | 65.24 | 38.23 | 64.40 |
| | SpareGPT | 2:4 (50%) | 75.24 | 65.67 | 65.10 | 59.76 | 34.04 | 59.96 |
| | Wanda | 2:4 (50%) | 75.46 | 63.54 | 63.41 | 60.14 | 31.91 | 58.89 |
| | SliceGPT | 25% | 75.30 | 66.61 | 69.42 | 63.55 | 35.67 | 62.11 |
| | SliceGPT | 30% | 74.97 | 65.04 | 68.15 | 63.55 | 34.64 | 61.27 |
| | SLEB | 20% | 76.93 | 67.40 | 70.62 | 61.99 | 37.37 | 62.86 |
| | Ours | 10% | 77.91 | 66.38 | 72.26 | 65.24 | 38.65 | 64.08 |
| | Ours | 20% | 76.71 | 65.98 | 70.83 | 64.94 | 37.63 | 63.22 |
| OPT-66B | Dense | - | 79.82 | 68.90 | 74.85 | 67.21 | 40.02 | 66.16 |
| | SpareGPT | 2:4 (50%) | 77.75 | 66.22 | 68.59 | 63.34 | 35.75 | 62.33 |
| | Wanda | 2:4 (50%) | 50.65 | 50.51 | 25.97 | 25.29 | 27.22 | 35.93 |
| | SliceGPT | 25% | 78.40 | 67.09 | 73.33 | 67.89 | 39.16 | 65.17 |
| | SliceGPT | 30% | 77.42 | 66.30 | 72.62 | 66.90 | 37.97 | 64.24 |
| | SLEB | 20% | 76.99 | 66.22 | 70.77 | 63.34 | 35.32 | 62.53 |
| | Ours | 10% | 79.65 | 68.43 | 74.82 | 66.84 | 40.10 | 65.97 |
| | Ours | 20% | 79.33 | 66.69 | 73.63 | 65.78 | 38.57 | 64.80 |
| LLaMA-2-7B | Dense | - | 79.11 | 69.06 | 75.99 | 74.58 | 46.25 | 69.00 |
| | SpareGPT | 2:4 (50%) | 72.14 | 64.96 | 58.93 | 60.90 | 34.22 | 58.23 |
| | Wanda | 2:4 (50%) | 70.84 | 62.27 | 55.33 | 57.58 | 31.91 | 55.59 |
| | SliceGPT | 25% | 66.87 | 63.38 | 54.16 | 58.46 | 34.56 | 55.49 |
| | SliceGPT | 30% | 63.55 | 61.33 | 49.62 | 51.77 | 31.23 | 51.50 |
| | SLEB | 20% | 73.07 | 58.96 | 62.47 | 56.48 | 33.02 | 56.80 |
| | Ours | 10% | 77.64 | 63.77 | 71.52 | 67.76 | 41.55 | 64.45 |
| | Ours | 20% | 76.71 | 61.17 | 63.82 | 62.42 | 37.20 | 60.26 |
| LLaMA-2-13B | Dense | - | 80.47 | 72.22 | 79.39 | 77.48 | 49.23 | 71.76 |
| | SpareGPT | 2:4 (50%) | 75.46 | 68.51 | 65.52 | 66.04 | 39.76 | 63.06 |
| | Wanda | 2:4 (50%) | 73.94 | 67.01 | 63.09 | 64.31 | 37.80 | 61.23 |
| | SliceGPT | 25% | 68.55 | 67.48 | 58.10 | 62.50 | 37.88 | 58.90 |
| | SliceGPT | 30% | 66.10 | 65.11 | 52.69 | 56.82 | 35.07 | 55.16 |
| | SLEB | 20% | 76.61 | 64.96 | 70.55 | 64.35 | 38.31 | 62.96 |
| | Ours | 10% | 79.33 | 71.98 | 77.57 | 74.33 | 46.59 | 69.96 |
| | Ours | 20% | 77.48 | 65.90 | 71.01 | 69.61 | 41.47 | 65.09 |
| LLaMA-2-70B | Dense | - | 82.70 | 77.98 | 83.84 | 80.98 | 57.34 | 76.57 |
| | SpareGPT | 2:4 (50%) | 80.03 | 76.56 | 76.09 | 76.94 | 49.74 | 71.87 |
| | Wanda | 2:4 (50%) | 80.30 | 74.66 | 79.22 | 76.35 | 51.19 | 72.34 |
| | SliceGPT | 25% | 74.92 | 75.37 | 68.84 | 77.90 | 51.71 | 69.75 |
| | SliceGPT | 30% | 72.31 | 73.56 | 63.69 | 73.40 | 47.61 | 66.11 |
| | SLEB | 20% | 80.14 | 72.93 | 77.21 | 75.38 | 48.38 | 70.81 |
| | Ours | 10% | 75.37 | 82.54 | 82.08 | 80.39 | 55.55 | 75.18 |
| | Ours | 20% | 80.74 | 73.80 | 79.30 | 77.69 | 52.47 | 72.80 |

