# OpenReview forum: "Rethinking the Impact of Heterogeneous Sublayers in Transformers"
_ICLR.cc/2025/Conference — ICLR 2025 Conference Withdrawn Submission_

### Official Review · Reviewer_6P8W · 2024-11-04

**Soundness:** 2
**Presentation:** 1
**Contribution:** 1
**Rating:** 1
**Confidence:** 5

**Summary:**

This paper discusses the impact of the two sub-layers in the transformer blocks of large language models. The authors introduce a novel perplexity-based metric to assess the importance of each heterogeneous sub-layer and progressively prune them according to their importance. Additionally, the paper takes into account the parameter counts of different sub-layers and normalizes them to fairly evaluate the importance of the heterogeneous sub-layers. The proposed method can maintain minimal performance loss while pruning 20% of the parameters.

**Strengths:**

1. This paper dives into the inner of transformers architecture, exploring the importance of different sub-layers.

2. The visualizations are well done, making it easier to understand.

**Weaknesses:**

1. This paper is extremely similar to a paper published in June this year[1]. Could you explain your essential innovation?

2. The experimental section seems to lack a comparison with some coarse-grained pruning methods, such as Shortened LLaMA[2], ShortGPT[3], and similar works I mentioned[1].

3. The proposed PPL-based iterative method in this paper appears inefficient, as it requires multiple iterations to calculate PPL. Moreover, given the lack of relevant experiments, it’s difficult to conclude that iteratively removing sub-layers is better than deciding which sub-layers to remove all at once[4].

4. The authors determine which sub-layers to prune based on a calibration dataset, but they do not analyze the impact of the number of data samples or the data domain.

5. The results presented in the Wanda paper show that 2:4 pruning can achieve a 1.24× speedup on LLaMA-7B, while the result in this paper is 1.10×. The paper mentions using 2 GPUs to benchmark LLaMA-2-70B on 2 NVIDIA A100 GPUs, which may introduce inaccuracies due to the unavoidable communication overhead. This could be a potential reason for the discrepancy in results.

6. The evaluation tasks seem somewhat simple. providing results on tasks such as MMLU, GSM8K, and HumanEval would greatly help demonstrate the effectiveness of the method.

7. It is recommended to highlight the best results in all tables, as this allows readers to quickly grasp the key outcomes.

8. opt has provided valuable insights for the community, but its performance does not seem to compare with current mainstream LLMs. Providing results for models like Mistral and LLaMA-3 would be more convincing.

9. There are some issues with the footer on page 5 and the header on page 6.

10. In Line 796, the PIQA result with 10% of the parameters pruned is worse than with 20% pruned, which is odd. Is this due to a data recording error?

References

[1] He S, Sun G, Shen Z, et al. What matters in transformers? not all attention is needed[J]. arXiv preprint arXiv:2406.15786, 2024.

[2] Kim B K, Kim G, Kim T H, et al. Shortened llama: A simple depth pruning for large language models[J]. arXiv preprint arXiv:2402.02834, 2024, 11.

[3] Men X, Xu M, Zhang Q, et al. Shortgpt: Layers in large language models are more redundant than you expect[J]. arXiv preprint arXiv:2403.03853, 2024.

[4] Gromov A, Tirumala K, Shapourian H, et al. The unreasonable ineffectiveness of the deeper layers[J]. arXiv preprint arXiv:2403.17887, 2024.

**Questions:**

Please refer to the questions and suggestions listed under Weaknesses.

---

### Official Review · Reviewer_LFZG · 2024-11-04

**Soundness:** 3
**Presentation:** 3
**Contribution:** 3
**Rating:** 5
**Confidence:** 4

**Summary:**

This paper presents a post-training layer pruning method to address the complexity of large language models (LLMs). It focuses on sublayers inside a transformer block, Multi-Head Attention, and Feed-Forward Network. Evaluating the importance of each sublayer with the proposed normalized relative impact factor metric removes sublayers that minimize the impact of the evaluation results in terms of perplexity. The experiments show that the methods reduce parameters by 20% while maintaining minimal performance loss.

**Strengths:**

1. The paper provides a detailed analysis of each layer's activation, including its output magnitude, input-output similarity, and importance. It also provides insight into the redundancy of each layer in LLMs, which is helpful for further improving other pruning methods.
2. The sublayer pruning method is neat and clean. It simply removes all the sublayers and does not require additional hardware support. More importantly, it can obtain relatively good performance compared to baselines.
3. The paper is well-written and easy to follow.

**Weaknesses:**

1. Lack of ablation study:
    1. The paper argues that MHAs are more redundant than FFNs, however there is no ablation study on pruning MHA-only vs. FFN-only under the same sparsity to support this conclusion.
    2. The paper proposed progressive layer search; there is no ablation study that demonstrates the effectiveness of the search method, for example, naive search (e.g., identifying all indexes in one round).

2. High complexity for progressive search: Progressive layer search requires iteratively examining all possible layers within a model. The complexity will grow with the increasing number of layers in LLMs, as shown in Table 5. The run time will increase, when pruning a larger model or with higher sparsity,

**Questions:**

1. Please see weakness.
2. What is the actual performance of LLMs after pruning? Although the `LM Evaluation Harness` evaluation looks good, sometimes the actual performance of the model might still be impacted in pratice. I am curious if the pruning method can still maintain reasonable responses. Could you please provide a response to the same query before and after pruning?

---

### Official Review · Reviewer_aakw · 2024-11-04

**Soundness:** 2
**Presentation:** 2
**Contribution:** 2
**Rating:** 3
**Confidence:** 4

**Summary:**

This paper revisits transformer architecture in large language models (LLMs) to offer an analysis of the impacts of specific sublayers like Multi-Head Attention (MHA) and Feed-Forward Network (FFN). It introduces a metric called the normalized relative impact factor, which guides a progressive, heterogeneous layer pruning technique that achieves a 20% reduction in parameters and a 37% increase in inference speed some performance loss.

**Strengths:**

- **Real-Device Efficiency in Pruning Techniques**: Developing pruning techniques that enhance real-device efficiency is a valuable direction to support the broader deployment of redundant LLM models.

- **Clarity and Intuitiveness of Proposed Approach**: The proposed approach is intuitive, and the overall presentation is clear and easy to understand.

**Weaknesses:**

- **Incremental Contribution Due to Existing Guidelines**: The guidelines discussed in the paper largely reflect established observations, making the contribution of this work relatively incremental. Specifically, Guideline 1, regarding the higher redundancy of intermediate layers, has been previously noted in [1,2], and Guideline 2, observing greater layer redundancy in large models, has been discussed in [1].

- **Definition of Relative Impact Factor (RI)**: The definition of the relative impact factor (RI) relies solely on perplexity. While the reasoning behind this metric is understandable, existing studies have highlighted that perplexity is not always a reliable evaluation metric [3,4]. It would be beneficial if the authors could elaborate on the rationale behind choosing this metric, as well as discuss its strengths and limitations.

- **Benchmarking Against Competitive Baselines**: It would be highly valuable if the authors could include additional competitive baselines, such as [1,5,6], for a more comprehensive comparison. Moreover, to better assess the pruned model’s performance on challenging tasks, it would be advantageous to evaluate the proposed approach against existing baselines on open-ended generation tasks and knowledge-intensive benchmarks, including MT-Bench [7], Alpaca-Eval [8], and MMLU [9].

[1] Kim, Bo-Kyeong, et al. "Shortened llama: A simple depth pruning for large language models." arXiv preprint arXiv:2402.02834 11 (2024).

[2] Yu, Zhongzhi, et al. "EDGE-LLM: Enabling Efficient Large Language Model Adaptation on Edge Devices via Layerwise Unified Compression and Adaptive Layer Tuning and Voting." arXiv preprint arXiv:2406.15758 (2024).

[3] Meister, Clara, and Ryan Cotterell. "Language model evaluation beyond perplexity." arXiv preprint arXiv:2106.00085 (2021).

[4] Wang, Yequan, et al. "Perplexity from plm is unreliable for evaluating text quality." arXiv preprint arXiv:2210.05892 (2022).

[5] Men, Xin, et al. "Shortgpt: Layers in large language models are more redundant than you expect." arXiv preprint arXiv:2403.03853 (2024).

[6] Dery, Lucio, et al. "Everybody prune now: Structured pruning of llms with only forward passes." arXiv preprint arXiv:2402.05406 (2024).

[7] Zheng, Lianmin, et al. "Judging llm-as-a-judge with mt-bench and chatbot arena." Advances in Neural Information Processing Systems 36 (2023): 46595-46623.

[8] Li, Xuechen, et al. AlpacaEval: An Automatic Evaluator of Instruction-following Models. GitHub, May 2023, https://github.com/tatsu-lab/alpaca_eval.

[9] Hendrycks, Dan, et al. "Measuring massive multitask language understanding." arXiv preprint arXiv:2009.03300 (2020).

**Questions:**

Please refer to the weakness section.

---

### Official Review · Reviewer_P46L · 2024-11-06

**Soundness:** 3
**Presentation:** 3
**Contribution:** 2
**Rating:** 5
**Confidence:** 5

**Summary:**

This works studies the importance of sublayers in pretrained LLM. A metric, normalized relative impact factor, is introduced to calcluate the importance of each sublayer and conduct progress prunning. Experiments demonstrate that the approach can lead to a 20% reduction in parameters and a 37% inference speedup, while maintaining minimal performance loss.

**Strengths:**

* The unraveled view of the transformer is intuitive and interesting. In addition, the analysis on the feature L2 norm for the residual connection and MHA/FFN also provides intriguing insights.
* The presentation is clear and easy to follow. Especially, the several guidelines provided are helpful for the community to develop new architectures.
* Experiments are done on different model scales, ranging from 6.7B to 70B.
* Real acceleration on GPU is profiled in order to better understand the benefit of layer dropping.

**Weaknesses:**

* Layer dropping has been studied in several existing literature like [1,2,3]. I found that the methods and conclusions involved in the current submission are quite similar to other literature. It is hard for me to understand what the novelty and unique contribution of the current submission is.
* This work emphasizes "progressive pruning". However, it is well known in pruning studies. In addition, I did not see any link between "progressive pruning" and "NRI". (One can do "progressive pruning" with other metrics as well.)
* Evaluation of the pruned model merely includes a few tasks like PIQA, Wino., Hella., ARC-e and ARC-c. I was wondering how the pruned models perform on more challenging tasks like MMLU, gsm8k, AGIEval and recall-intensive tasks. It is important to comprehensively understand the impact of layer dropping.

[1] Men, Xin, et al. "Shortgpt: Layers in large language models are more redundant than you expect." arXiv preprint arXiv:2403.03853 (2024).
[2] Gromov, Andrey, et al. "The unreasonable ineffectiveness of the deeper layers." arXiv preprint arXiv:2403.17887 (2024).
[3] Siddiqui, Shoaib Ahmed, et al. "A deeper look at depth pruning of LLMs." arXiv preprint arXiv:2407.16286 (2024).

**Questions:**

see weakness

---

### Note · Authors · 2024-11-15

I have read and agree with the venue's withdrawal policy on behalf of myself and my co-authors.